# Universal Weakly Supervised Segmentation by Pixel-to-Segment Contrastive Learning

**Tsung-Wei Ke**  **Jyh-Jing Hwang**  **Stella X. Yu**
UC Berkeley / ICSI
{twke,jyh,stellayu}@berkeley.edu

## Abstract

Weakly supervised segmentation requires assigning a label to every pixel based on training instances with partial annotations such as image-level tags, object bounding boxes, labeled points and scribbles. This task is challenging, as coarse annotations (*tags*, *boxes*) lack precise pixel localization whereas sparse annotations (*points*, *scribbles*) lack broad region coverage. Existing methods tackle these two types of weak supervision differently: Class activation maps are used to localize coarse labels and iteratively refine the segmentation model, whereas conditional random fields are used to propagate sparse labels to the entire image.

We formulate weakly supervised segmentation as a semi-supervised metric learning problem, where pixels of the same (different) semantics need to be mapped to the same (distinctive) features. We propose 4 types of contrastive relationships between pixels and segments in the feature space, capturing low-level image similarity, semantic annotation, co-occurrence, and feature affinity. They act as priors; the pixel-wise feature can be learned from training images with any partial annotations in a data-driven fashion. In particular, unlabeled pixels in training images participate not only in data-driven grouping within each image, but also in discriminative feature learning *within* and *across* images. We deliver a universal weakly supervised segmenter with significant gains on Pascal VOC and DensePose. Our code is publicly available at https://github.com/twke18/SPML.

## 1 Introduction

Consider the task of learning a semantic segmenter given sparsely labeled training images (Fig. 1): Each body part is labeled with a single seed pixel and the task is to segment out the entire person

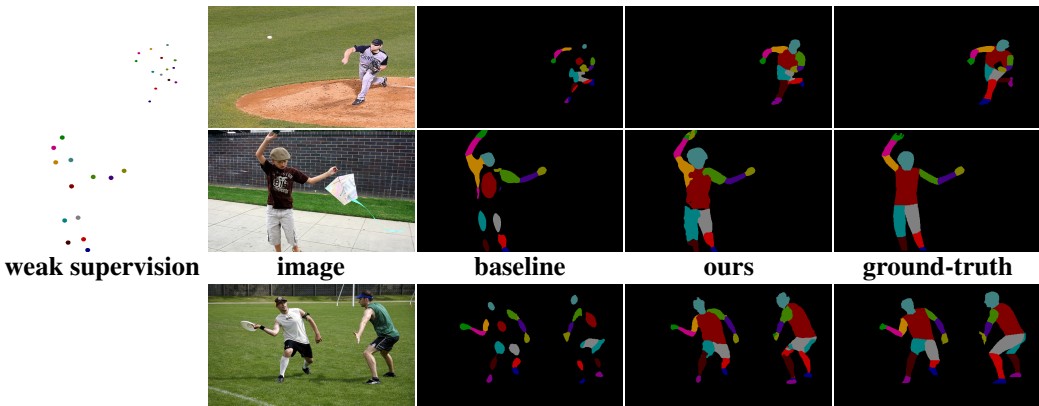

weak supervision   image   baseline   ours   ground-truth

Figure 1: Our task learns a segmenter given partially labeled training images and applies it to test images. A common baseline is to propagate labels within an image based on feature similarity. We model it as semi-supervised metric learning and learn the pixel-wise feature by contrasting it within and across images. Our results are fuller and more accurate, approaching the ground-truth.

| image | image tags | bounding boxes | labeled points | scribbles |
|---|---|---|---|---|
| | **Person** **Motorbike** | | | |
| **SOTA methods** | CAM + refine | box-wise CAM | CRF loss | CRF loss |
| **our method** | single pixel-to-segment contrastive learning loss formulation | | | |
| **our relative gain** | +8.6% | +4.7% | +24.7% | +1.4% |

Figure 2: We propose a unified framework for weakly supervised semantic segmentation with different types of annotations. We demonstrate consistent performance gains compared to the state-of-the-art (SOTA) methods: Chang et al. (2020) for image tags, Song et al. (2019) for bounding boxes, and Tang et al. (2018b) for points and scribbles. For tags and boxes, Class Activation Maps (CAM) (Zhou et al., 2016) are often used to localize semantics as an initial mask and iteratively refine the segmentation model, whereas for labeled points and scribbles, Conditional Random Fields (CRF) are used to propagate semantic labels to unlabeled regions based on low-level image similarity.

by individual body parts, even though the ground-truth segmentation is not known during training. This task is challenging, as not only a single body part could contain several visually distinctive areas (*e.g., head* consists of *eyes*, *nose*, *mouth*, *beard*), but two adjacent body parts could also have the same visual appearance (*e.g., upper arm*, *lower arm*, and *hand* have the same skin appearance). Once the segmenter is learned, it can be applied to a test image without any annotations.

This task belongs to a family of weakly supervised segmentation problems, the goal of which is to assign a label to each pixel despite that only partial supervision is available during training. It addresses the practical issue of learning segmentation from minimum annotations. Such weak supervision takes many forms, e.g., image tags (Kolesnikov & Lampert, 2016; Ahn & Kwak, 2018; Huang et al., 2018; Lee et al., 2019), bounding boxes (Dai et al., 2015; Khoreva et al., 2017; Song et al., 2019), keypoints (Bearman et al., 2016), and scribbles (Lin et al., 2016; Tang et al., 2018a;b). Tags and boxes are coarse annotations that lack precise pixel localization whereas points and scribbles are sparse annotations that lack broad region coverage.

Weakly supervised semantic segmentation can be regarded as a semi-supervised pixel classification problem: Some pixels or pixel sets have labels, most don't, and the key is how to propagate and refine annotations from coarsely and sparsely labeled pixels to unlabeled pixels.

Existing methods tackle two types of weak supervision differently: Class Activation Maps (CAM) (Zhou et al., 2016) are used to localize coarse labels, generate pseudo pixel-wise labels, and iteratively refine the segmentation model, whereas Conditional Random Fields (CRF) (Krähenbühl & Koltun, 2011) are used to propagate sparse labels to the entire image. These ideas can be incorporated as an additional unsupervised loss on the feature learned for segmentation (Tang et al., 2018b): While labeled pixels receive supervision, unlabeled pixels in different segments shall have distinctive feature representations.

We propose a *Semi-supervised Pixel-wise Metric Learning* (SPML) model that can handle all these weak supervision varieties with a single pixel-to-segment contrastive learning formulation (Fig. 2). Instead of classifying pixels, our metric learning model learns a pixel-wise feature embedding based on common grouping relationships that can be derived from any form of weak supervision.

Our key insight is to integrate unlabeled pixels into both supervised labeling and discriminative feature learning. They shall participate not only in data-driven grouping within each image, but also in discriminative feature learning *within* and more importantly *across* images. Intuitively, labeled pixels receive supervision not only for themselves, but also for their surround pixels that share visual similarity. On the other hand, unlabeled pixels are not just passively brought into discriminative learning induced by sparsely labeled pixels, they themselves are organized based on bottom-up grouping cues (such as grouping by color similarity and separation by strong contours). When they are examined *across* images, repeated patterns of frequent occurrences would also form a cluster that demand active discrimination from other patterns.

We capture the above insight in a single pixel-wise metric learning objective for segmentation, the goal of which is to map each pixel into a point in the feature space so that pixels in the same (different) semantic groups are close (far) in the feature space. Our model extends SegSort (Hwang et al., 2019) from its fully supervised and unsupervised segmentation settings to a universal weakly-supervised segmentation setting. With a single consistent feature learning criterion, such a model sorts pixels discriminatively within individual images and sorts segment clusters discriminatively across images, both steps minimizing the same feature discrimination loss.

Our experiments on Pascal VOC (Everingham et al., 2010) and DensePose (Alp Güler et al., 2018) demonstrate consistent gains over the state-of-the-art (SOTA), and the gain is substantial especially for the sparsest keypoint supervision.

## 2 RELATED WORK

**Semi-supervised learning.** Weston et al. (2012) treats it as a joint learning problem with both labeled and unlabeled data. One way is to capture the underlying structure of unlabeled data with generative models (Kingma et al., 2014; Rasmus et al., 2015). Another way is to regularize feature learning through a consistency loss, *e.g.,* adversarial ensembling (Miyato et al., 2018), imitation learning and distillation (Tarvainen & Valpola, 2017), cross-view ensembling (Clark et al., 2018). These methods are most related to transductive learning (Joachims, 2003; Zhou et al., 2004; Fergus et al., 2009; Liu et al., 2019), where labels are propagated to unlabeled data via clustering in the pre-trained feature space. Our work does transductive learning in an adaptively learned feature space.

**Weakly-supervised semantic segmentation.** Partial annotations include scribbles (Lin et al., 2016; Tang et al., 2018a;b; Wang et al., 2019), bounding boxes (Dai et al., 2015; Khoreva et al., 2017; Song et al., 2019), points (Bearman et al., 2016), or image tags (Papandreou et al., 2015; Kolesnikov & Lampert, 2016; Ahn & Kwak, 2018; Huang et al., 2018; Li et al., 2018; Lee et al., 2019; Shimoda & Yanai, 2019; Zhang et al., 2019; Yao & Gong, 2020; Chang et al., 2020; Araslanov & Roth, 2020; Wang et al., 2020; Fan et al., 2020; Sun et al., 2020). Xu et al. (2015) formulates all types of weak supervision as linear constraints on a SVM. Papandreou et al. (2015) bootstraps segmentation predictions via EM-optimization. Recent works (Lin et al., 2016; Kolesnikov & Lampert, 2016; Pathak et al., 2015) typically use CAM (Zhou et al., 2016) to obtain an initial dense mask and then train a model iteratively. GAIN (Li et al., 2018) utilizes image tags or bounding boxes to refine these class-specific activation maps. Sun et al. (2020) considers within-image relationships and explores the idea of co-segmentation. Fan et al. (2020) estimates the foreground and background for each category, with which the network learns to generate more precise CAMs. Regularization is enforced at either the image level (Lin et al., 2016; Kolesnikov & Lampert, 2016; Pathak et al., 2015) or the feature level (Tang et al., 2018a;b) to produce better dense masks. We incorporate this concept into adaptive feature learning and train the model only once. All types of weak annotations are dealt with in a single contrastive learning framework.

**Non-parametric segmentation.** Prior to deep learning, non-parametric models (Russell et al., 2009; Tighe & Lazebnik, 2010; Liu et al., 2011) usually use designed features with statistical or graphical models to segment images. Recently, inspired by non-parametric models for recognition (Wu et al., 2018b;a), SegSort (Hwang et al., 2019) captures pixel-to-segment relationships via a pixel-wise embedding and develops the first deep non-parametric semantic segmentation for supervised and unsupervised settings. Building upon SegSort, our work has the flexibility of a non-parametric model at capturing data relationships and modeling subclusters within a category.

## 3 SEMI-SUPERVISED PIXEL-WISE METRIC LEARNING METHOD

Metric learning develops a feature representation based on data grouping and separation cues. Our method (Fig. 3) segments an image by learning a pixel-wise embedding with a contrastive loss between pixels and segments: For each pixel $i$, we learn a latent feature $\phi(i)$ such that $i$ is close to its positive segments (exemplars) and far from its negative ones in that feature space.

In the fully supervised setting, we can define pixel $i$'s positive and negative sets, denoted by $\mathcal{C}^+$ and $\mathcal{C}^-$ respectively, as pixels in the same (different) category. However, this idea is not applicable

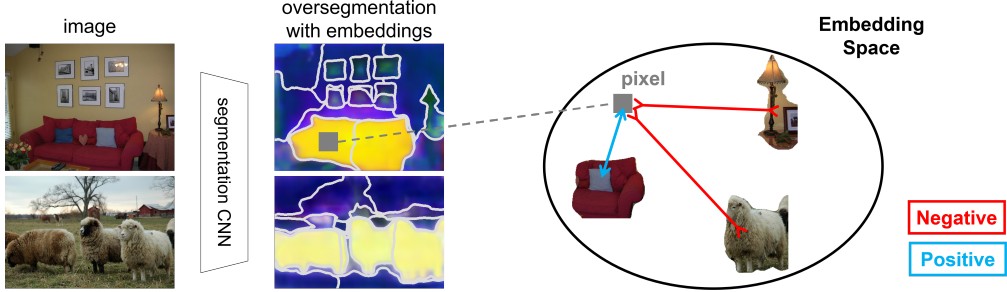

Figure 3: Overall method diagram. We develop pixel-wise embeddings with contrastive learning between pixels and segments. We derive various forms of positive and negative segments for each pixel. Our goal is to attract (blue inward arrows) the pixel with positive segments, while repelling (red outward arrows) it from negative segments in the feature space.

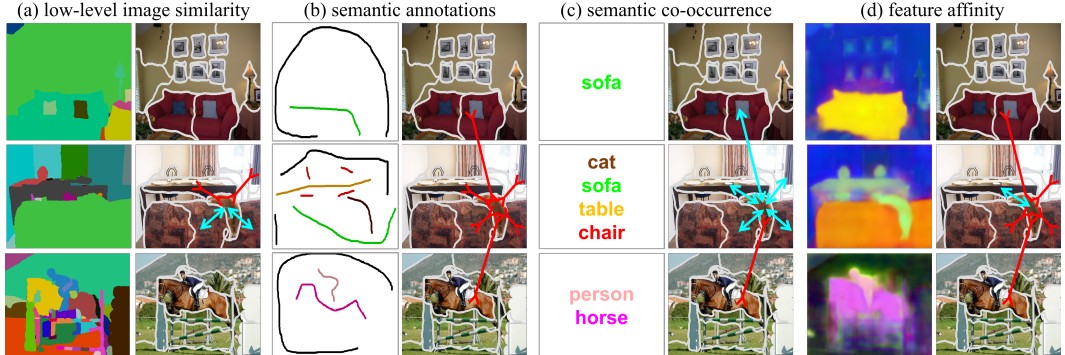

Figure 4: Four types of pixel-to-segment attraction and repulsion relationships. A pixel is attracted to (repelled by) segments: **a)** of similar (different) visual appearances such as color or texture, **b)** of the same (different) class labels, **c)** in images with common (distinctive) labels, **d)** of nearby (far-away) feature embeddings. They form different positive and negative sets.

to weakly- or un-supervised settings where the label is not available on every pixel. In the labeled points setting, $\mathcal{C}^+$ and $\mathcal{C}^-$ would only contain a few exemplars according to the sparse pixel labels.

Our basic idea is to enlarge the sets of $\mathcal{C}^+$ and $\mathcal{C}^-$ to improve the feature learning efficacy. By exploring different relationships and assumptions in the image data, we are able to generate abundant positive and negative segments for any pixel at the same time, providing more supervision in the latent feature space. We propose four types of relationships between pixels and segments (Fig. 4):

1. **Low-level image similarity**: We impose a spatial smoothness prior on the pixel-wise feature to keep pixels together in visually coherent regions. The segment pixel $i$ belongs to based on low-level image cues is a positive segment to pixel $i$; any other segments are negative ones.

2. **Semantic annotation**: We expand the semantics from labeled points and scribbles to pseudo-labels inferred from image- or box-wise CAM. The label of a segment can be estimated by majority vote among pixels; if it is the same as pixel $i$'s, the segment is a positive segment to $i$.

3. **Semantic co-occurrence**: We expand the semantics by assuming that pixels in similar semantic contexts tend to be grouped together. If a segment appears in an image that shares any of the semantic classes as pixel $i$'s image, it is a positive segment to $i$ and otherwise a negative one.

4. **Feature affinity**: We impose a featural smoothness prior assuming that pixels and segments of the same semantics form a cluster in the feature space. We propagate the semantics within and across images from pixel $i$ to its closest segment $s$ in the feature space.

## 3.1 Pixel-to-Segment Contrastive Grouping Relationships

Our goal is to propagate known semantics from labeled data $\mathcal{C}$ to unlabeled data $\mathcal{U}$ with the afore-mentioned priors. $\mathcal{C}$ and $\mathcal{U}$ denote the sets of segment indices respectively. We detail how to augment positive / negative segment sets using both $\mathcal{C}$ and $\mathcal{U}$ for each type of relationships (Fig. 4).

**Low-level image similarity.** To propagate labels within visually coherent regions, we generate a low-level over-segmentation. Following SegSort (Hwang et al., 2019), we use the HED contour detector (Xie & Tu, 2015) (pre-trained on BSDS500 dataset (Arbelaez et al., 2010)) and gPb-owt-ucm (Arbelaez et al., 2010) to generate a segmentation without semantic information. We define $i$'s positive and negative segments as $i$'s own segment and all the other segments, denoted as $\mathcal{V}^+$ and $\mathcal{V}^-$ respectively. We only consider segments in the same image as pixel $i$'s. We align the contour-based over-segmentations with segmentations generated by K-Means clustering as in SegSort.

**Semantic annotation.** Image tags and bounding boxes do not provide pixel-wise localization. We derive pseudo labels from image- or box-wise CAM and align them with oversegmentations induced by the pixel-wise feature. Pixel $i$'s positive (negative) segments are the ones with the same (different) semantic category, denoted by $\mathcal{C}^+$ and $\mathcal{C}^-$ respectively. We ignore all the unlabeled segments.

**Semantic co-occurrence.** Semantic context characterizes the co-occurrences of different objects, which can be used as a prior to group and separate pixels. We define semantic context as the union of object classes in each image. Even without the pixel-wise localization of semantic labels, we can leverage semantic context to impose global regularization on the latent feature: The feature should separate images without any overlapping object categories.

Let $\mathcal{O}^+$ ($\mathcal{O}^-$) denote the set of segments in images with (without) overlapping categories as pixel $i$'s image. That is, if the image of pixel $i$ and another image share any semantic labels (Fig. 4c: {*cat, sofa, table, chair*} for the pixel in the Row 2 image vs. {*sofa*} for the Row 1 image), then all the segments from that image are positive segments to $i$ and included in $\mathcal{O}^+$; otherwise they are considered negative segments in $\mathcal{O}^-$ (Fig. 4c: all the segments in the Row 3 image). In particular, all the segments in pixel $i$'s image are in $\mathcal{O}^+$ of $i$. This semantic context relationship does not require localized annotations yet imposes regularization on pixel feature learning.

**Feature affinity.** Our goal is to learn a pixel-wise feature that indicates semantic segmentation. It is thus reasonable to assume that pixels and segments of the same semantics form a cluster in the feature space, and we reinforce such clusters with a featural smoothness prior: We find nearest neighbours in the feature space and propagate labels accordingly.

Specifically, we assign a semantic label to each unlabeled segment by finding its nearest labeled segment in the feature space. We denote this expanded labeled set by $\hat{\mathcal{C}}$. For pixel $i$, we define its positive (negative) segment set $\hat{\mathcal{C}}^+$ ($\hat{\mathcal{C}}^-$) according to whether a segment has the same label as $i$.

Our feature affinity relationship works best when: 1) the original labeled set is large enough to cover the feature space, 2) the labeled segments are distributed uniformly in the feature space, and 3) the pixel-wise feature already encodes certain semantic information. We thus only apply to DensePose keypoint annotations in our experiments, where each body part is annotated by a point.

## 3.2 Pixel-wise Metric Learning Loss

SegSort (Hwang et al., 2019) is an end-to-end segmentation model that generates a pixel-wise feature map and a resulting segmentation. Assuming independent normal distributions for individual segments, SegSort seeks a maximum likelihood estimation of the feature mapping, so that the feature induced partitioning in the image and clustering across images provide maximum discrimination among segments. During inference, the segment label is predicted by K-Nearest Neighbor retrievals.

The feature induced partitioning in each image is calculated via spherical K-Means clustering (Banerjee et al., 2005). Let $\boldsymbol{e}_i$ denote the feature vector at pixel $i$, which contains the mapped feature $\phi(i)$ and $i$'s spatial coordinates. Let $z_i$ denote the index of the segment that pixel $i$ belongs to, $\boldsymbol{R}_s$ the set of pixels in segment $s$, and $\boldsymbol{\mu}_s$ the segment feature calculated as the spherical cluster centroid of segment $s$. In the Expectation-Maximization (EM) procedure for spherical K-means, the E-step calculates the most likely segment pixel $i$ belongs to: $z_i = \arg\max_s \boldsymbol{\mu}_s' \boldsymbol{e}_i$, and the M-Step updates the segment feature as the mean pixel-wise feature: $\boldsymbol{\mu}_s = \frac{\sum_{i \in \boldsymbol{R}_s} \boldsymbol{e}_i}{\| \sum_{i \in \boldsymbol{R}_s} \boldsymbol{e}_i \|}$.

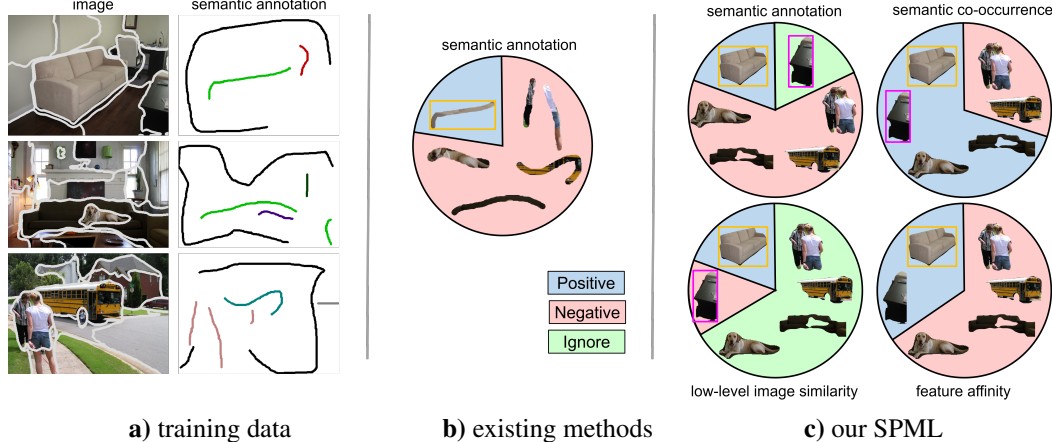

Figure 5: Our method uses labeled and unlabeled portions of the training data more extensively. **a)** Training images and their labeled scribbles are sparse and incomplete. **b)** Existing methods train a pixel-wise classifier using only labeled pixels and propagate labels within each image. **c)** Our method leverages *four* types of pixel-to-segment semantic relationships to augment the labeled sets, includes unlabeled pixels (fuller segments than just thin scribbles) and unlabeled segments (e.g. *desk* outlined in magenta), forms dynamic contrastive relationships between segments (e.g. the *desk* can be positive, negative, or to be ignored to the *sofa* in different relations.

Let $s$ denote the resulting segment that pixel $i$ belongs to per spherical clustering. The posterior probability of pixel $i$ in segment $s$ can be evaluated over the set of all segments $S$ as:

$$p(z_i = s|\boldsymbol{e}_i, \boldsymbol{\mu}) = \frac{\exp(\kappa \boldsymbol{\mu}'_s \boldsymbol{e}_i)}{\sum_{t \in S} \exp(\kappa \boldsymbol{\mu}'_t \boldsymbol{e}_i)} \tag{1}$$

where $\kappa$ is a concentration hyper-parameter. SegSort minimizes the negative log-likelihood loss:

$$L_{\text{SegSort}}(i) = -\log p(z_i = s|\boldsymbol{e}_i, \boldsymbol{\mu}) = -\log \frac{\exp(\kappa \boldsymbol{\mu}'_s \boldsymbol{e}_i)}{\sum_{t \in S} \exp(\kappa \boldsymbol{\mu}'_t \boldsymbol{e}_i)}. \tag{2}$$

SegSort adopts soft neighborhood assignment (Goldberger et al., 2005) to further strengthen the grouping of same-category segments. Let $\mathcal{C}^+$ ($\mathcal{C}^-$) denote the index set of segments in the same (different) category as pixel $i$ except $s$ – the segment $i$ belongs to. We have:

$$L_{\text{SegSort}+}(i, \mathcal{C}^+, \mathcal{C}^-) = -\log \sum_{t \in \mathcal{C}^+} p(z_i = t|\boldsymbol{e}_i, \boldsymbol{\mu}) = -\log \frac{\sum_{t \in \mathcal{C}^+} \exp(\kappa \boldsymbol{\mu}'_t \boldsymbol{e}_i)}{\sum_{t \in \mathcal{C}^+ \cup \mathcal{C}^-} \exp(\kappa \boldsymbol{\mu}'_t \boldsymbol{e}_i)}. \tag{3}$$

For our weakly supervised segmentation, the total pixel-to-segment contrastive loss for pixel $i$ consists of 4 terms, one for each of the 4 pixel-to-segment attraction and repulsion relationships:

$$L(i) = \lambda_I L_{\text{SegSort}+}(i, \mathcal{V}^+, \mathcal{V}^-) + \lambda_C L_{\text{SegSort}+}(i, \mathcal{C}^+, \mathcal{C}^-)$$
$$+ \lambda_O L_{\text{SegSort}+}(i, \mathcal{O}^+, \mathcal{O}^-) + \lambda_A L_{\text{SegSort}+}(i, \hat{\mathcal{C}}^+, \hat{\mathcal{C}}^-), \tag{4}$$

where $\lambda_C = 1$. Fig. 5 shows how our metric learning method utilizes labeled and unlabeled pixels and segments more extensively than existing classification methods: Our pseudo-labeled sets are fuller than labeled thin scribbles and include unlabeled segments; there are 3 more relationships other than semantic annotations; our segments participate in contrastive learning with dynamic roles in different relations. By easily integrating a full range of pixel-to-segment attraction and repulsion relationships from low-level image similarity to mid-level feature affinity, and to high-level semantic co-occurrence, we go far beyond the direct supervision from semantic annotations.

# 4 EXPERIMENTS ON PASCAL VOC AND DENSEPOSE

**Datasets. Pascal VOC 2012** Everingham et al. (2010) includes 20 object categories and one background class. Following Chen et al. (2017), we use the augmented training set with 10,582 images and validation set with 1,449 images. We use the scribble annotations provided by Lin et al.

| Dataset | Annotation | $\lambda_I$ | $\lambda_O$ | $\lambda_A$ |
|---------|------------|-------------|-------------|-------------|
| Pascal | image tags | 0.3 | 1.0 | 0.0 |
| | boxes | 0.3 | 1.0 | 0.0 |
| | points | 1.0 | 1.0 | 0.0 |
| | scribbles | 0.1 | 0.5 | 0.0 |
| DensePose | points | 0.1 | 0.0 | 0.5 |

Table 1: Hyper-parameters for different types of annotations on Pascal and DensePose dataset.

(2016) for training. **DensePose** (Alp Güler et al., 2018) is a human pose parsing dataset based on MSCOCO (Lin et al., 2014). The dataset is annotated with 14 body part classes. We extract the keypoints from the center of each part segmentation. The training set includes 26,437 images and we use minival2014 set for testing, which includes 1,508 images. See Appendix for more details.

**Architecture, training and testing.** For all the experiments on PASCAL VOC, we base our architecture on DeepLab (Chen et al., 2017) with ResNet101 (He et al., 2016) as the backbone network. For the experiments on DensePose, we adopt PSPNet (Zhao et al., 2017) as the backbone network. Our models are pre-trained on ImageNet (Deng et al., 2009) dataset. See Appendix for details on our inference procedure and hyper-parameter selection for training and testing.

For each type of annotations and dataset, we formulate four types of pixel-to-segment contrastive relationships and jointly optimize them in a single pixel-wise metric learning framework (Fig. 3). Table 1 shows the data-driven selection of hyperparameters $\lambda_I$, $\lambda_O$ and $\lambda_A$ for different task settings.

**Pascal: Image tag annotations.** Table 2 shows that, without using additional saliency labels, our method outperforms existing methods with saliency by $4.4\%$, and those without saliency by $5.1\%$.

**Pascal: Bounding box annotations.** Table 2 shows that, with the same DeepLab/ResNet101 backbone network, our method outperforms existing methods by $3.2\%$.

| Pascal: Image tags | Saliency | *val* | *test* |
|--------------------|----------|-------|--------|
| Huang et al. (2018) | ✓ | 61.4 | 63.2 |
| Lee et al. (2019) | ✓ | 64.9 | 65.3 |
| Zhang et al. (2019) | - | 66.3 | 66.5 |
| Yao & Gong (2020) | ✓ | 67.1 | 67.2 |
| Chang et al. (2020) | - | 66.1 | 65.9 |
| Our SPML | - | **69.5** | **71.6** |

| Pascal: Bounding boxes | *val* | *test* |
|------------------------|-------|--------|
| Khoreva et al. (2017) | 69.4 | - |
| Song et al. (2019) | 70.2 | - |
| Our SPML | **73.5** | **74.7** |

Table 2: Pascal VOC 2012 dataset with image tag (left) and bounding box (right) annotations.

| Pascal: Scribbles | CRF | Full | Weak | WvF |
|-------------------|-----|------|------|-----|
| Tang et al. (2018a) | | 75.6 | 72.8 | 96.3 |
| Tang et al. (2018a) | ✓ | 76.8 | 74.5 | 97.0 |
| Tang et al. (2018b) | | 75.6 | 73.0 | 96.6 |
| Tang et al. (2018b) | ✓ | 76.8 | 75.0 | 97.7 |
| Wang et al. (2019) | | 75.6 | 73.2 | 96.8 |
| Wang et al. (2019) | ✓ | 76.8 | 76.0 | 99.0 |
| Our SPML | | 76.1 | **74.2** | **97.5** |
| Our SPML | ✓ | 77.3 | 76.1 | 98.4 |

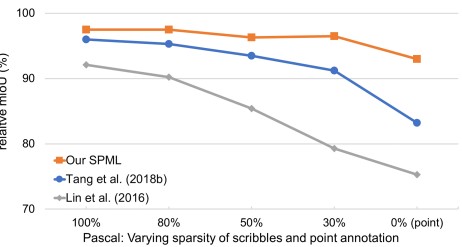

Table 3: Pascal VOC 2012 dataset using scribble annotations. **Left**: mIoU on validataion (white) and test (gray) set. **WvF** denotes relative mIoU w.r.t full supervision. **Right**: Relative mIoU performance w.r.t full supervision on different lengths of scribbles.

**Pascal: Scribble annotations.** Table 3 shows that, our method consistently delivers the best performance among methods without or with CRF post-processing. We get $74.2\%$ ($76.1\%$) mIoU, achieving $97.5\%$ ($98.4\%$) of full supervision performance in these two categories respectively.

**Pascal: Varying sparsity of scribble and point annotations.** Exploiting metric learning with different relationships in the data frees us from the classification framework and delivers a more

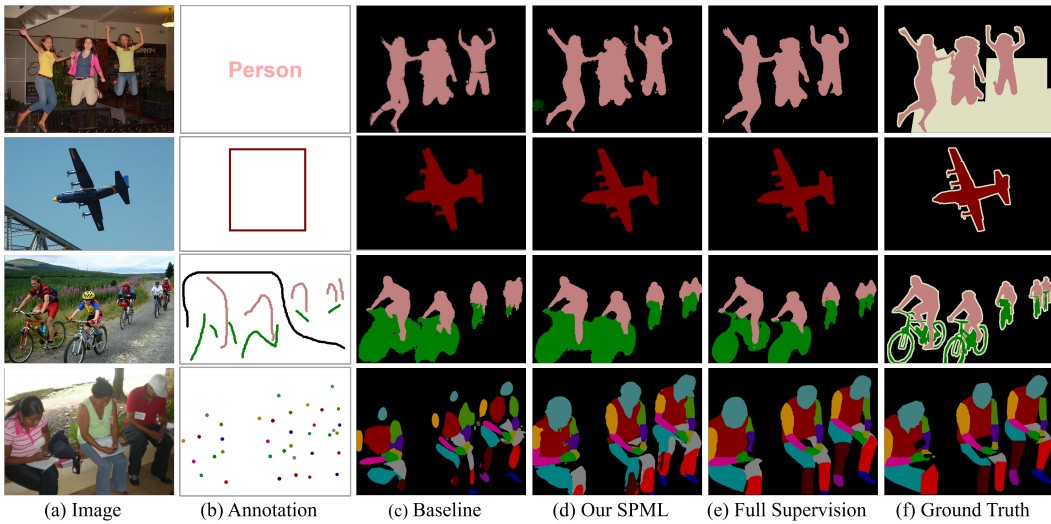

|  |  |  |  |  |  |
|---|---|---|---|---|---|
| (a) Image | (b) Annotation | (c) Baseline | (d) Our SPML | (e) Full Supervision | (f) Ground Truth |

Figure 6: Our results on Pascal and DensePose under various weak supervision settings are consistently better aligned with region boundaries and visually closer to fully supervised counterparts.

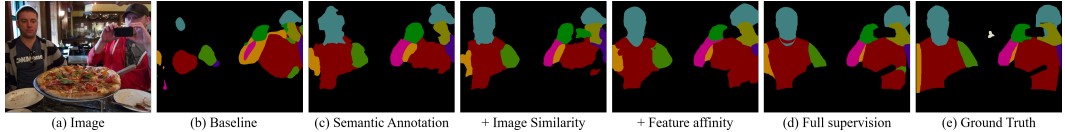

|  |  |  |  |  |  |  |
|---|---|---|---|---|---|---|
| (a) Image | (b) Baseline | (c) Semantic Annotation | + Image Similarity | + Feature affinity | (d) Full supervision | (e) Ground Truth |

Figure 7: Our segmentation results get better with more types of regularizations. We compare visual results by adding more regularizations. As we introduce more relationships for regularization, we observe significant improvement and our results are visually closer to fully supervised counterparts.

powerful approach that requires fewer annotations. Table 3 shows that, as we shorten the length of scribbles from $100\%, 80\%, 50\%, 30\%$ to $0\%$ (points), we reach $97.5\%, 97.5\%, 96.3\%, 96.5\%$ and $93.7\%$ of full supervision performance. Compared to the full scribble annotations, our accuracy only drops $3.7\%$ with point labels and is significantly better than the baseline.

| DensePose: Points | mIoU | WvF |
|---|---|---|
| Tang et al. (2018b) | 31.3 | 51.9 |
| Our SPML | **44.2** | **77.1** |

Table 4: DensePose minival 2014 set.

**DensePose: Point annotations.** We train our baseline using the code released by Tang et al. (2018b). Table 4 shows that, our method without CRF post-processing outperforms the baseline by $12.9\%$ mIoU, reaching $77.1\%$ of full supervision performance with only point supervision.

**Visual quality and ablation study.** Fig. 6 shows that our results are better aligned with region boundaries and visually closer to fully-supervised counterparts. Fig. 7 shows that our results improve significantly with different relationships for more regularization. See Appendix for more details and ablation studies.

**Summary.** We propose a novel weakly-supervised semantic segmentation method via Semi-supervised Pixel-wise Metric Learning, based on four common types of pixel-to-segment attraction and repulsion relationships. It is universally applicable to various weak supervision settings, whether the training images are coarsely annotated by image tags or bounding boxes, or sparsely annotated by keypoints or scribbles. Our results on PASCAL VOC and DensePose show consistent and substantial gains over SOTA, especially for the sparsest keypoint supervision.

**Acknowledgements.** This work was supported, in part, by Berkeley Deep Drive and Berkeley AI Research Commons with Facebook. This work used the Extreme Science and Engineering Discovery Environment (XSEDE), which is supported by National Science Foundation grant number ACI-1548562. Specifically, it used the Bridges system, which is supported by NSF award number ACI-1445606, at the Pittsburgh Supercomputing Center (PSC).

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

# A APPENDIX

We propose a single pixel-to-segment contrastive learning loss formulation for weakly supervised semantic segmentation. We explore different types of visual relationships to group and separate pixels *within* and *across* images. We demonstrate state-of-the-art performance using our proposed method with different types of annotations. Here, we include more details on the following aspects:

- We present the visual results of our method in A.1.

- We showcase the semantic cues generated by CAM for image tag and bounding box annotations in A.2.

- We illustrate the data pre-processing used for DensePose dataset in A.3.

- We describe the details of our experimental settings, hyper-parameters and inference procedure in A.4.

- We present the ablation study regarding hyper-parameters in A.5.

- We present mIoU performance with varying sparsity of scribble annotations on Pascal dataset in A.6.

- We present per-category results with Pascal and DensePose dataset in A.7.

## A.1 VISUALIZATION

We present the visual results on VOC (with image tags, bounding boxes and scribbles) and DensePose (with keypoints) dataset in figure 8. We observe that our segmentation results are better aligned with image boundary. When visual evidence is prominent, our weakly-supervised results are even better than the fully-supervised counterpart.

We then demonstrate the efficacy of each visual relationship in figure 9. By adding **semantic annotation**, **low-level image similarity** and **feature affinity** progressively, we observe consistent improvement of our results. The predicted segmentation becomes more coherent and better aligned with image boundary. We lastly showcase that our method implicitly encodes semantic contexts. In figure 10, We observe that retrieved segments appear in the similar semantic context as the query segments. For examples, given a bottle next to a desktop, our model retrieves bottles also next to a desktop; a set of sofas in a living room can be retrieved using one sofa query example; screens of a desktop can also be retrieved likewise.

## A.2 SEMANTIC ANNOTATIONS

Since image tag and bounding box annotations do not provide any of precisely localized semantic information, we adopt CAM (Zhou et al., 2016) to produce localized semantic cues. Without using additional saliency labels, we use the classifier trained by Wang et al. (2020) to generate CAM. Let $\mathcal{M}_c$ be the activation map of class $c$.

For image tag annotations, we follow Ahn & Kwak (2018) to normalize $\mathcal{M}_c$ of the entire image within the range between 0 and 1, where $\mathcal{M}_c = \frac{\mathcal{M}_c}{\max_c \mathcal{M}_c}$. The background confidence $\mathcal{M}_{bg}$ can then be estimated by $\mathcal{M}_{bg} = (1 - \max_c \mathcal{M}_c)^\alpha$, where $\alpha$ is the hyper-parameter adjusting background confidence. In our experiments, we set $\alpha$ to 6 and confidence threshold to 0.2. The low-confidence pixels are considered as unlabeled regions.

For bounding box annotations, we simply normalize the CAM logits within each bounding box to the range between 0 and 1. We then set confidence threshold to 0.5 for selecting foreground pixels and unlabeled regions. We restrict all the regions outside bounding boxes as "background". See figure 11 for more visual examples.

## A.3 DATA PRE-PROCESSING FOR DENSEPOSE DATASET

We next illustrate our pre-processing to generate training labels given keypoint annotations in DensePose dataset. As shown in figure 12, we first assume a Gaussian heat map from every keypoint. By

thresholding, we derive 3 regions from every Gaussian blob: labelled, unknown and background region. In labelled region, pixels are annotated as each body part. We then propagate labels, including background class, to pixels in the unknown region. The $std$ of Gaussian heat map is estimated from instance size, and we use ground-truth information in our paper.

### A.4 HYPER-PARAMETERS AND EXPERIMENTAL SETUP

**Architecture and training.** For all the experiments on VOC, we base our architecture as DeepLab (Chen et al., 2017) with ResNet101 (He et al., 2016) as backbone network. For the experiments on DensePose dataset, we adopt PSPNet (Zhao et al., 2017) as backbone network. We only use models pre-trained on ImageNet (Deng et al., 2009) dataset.

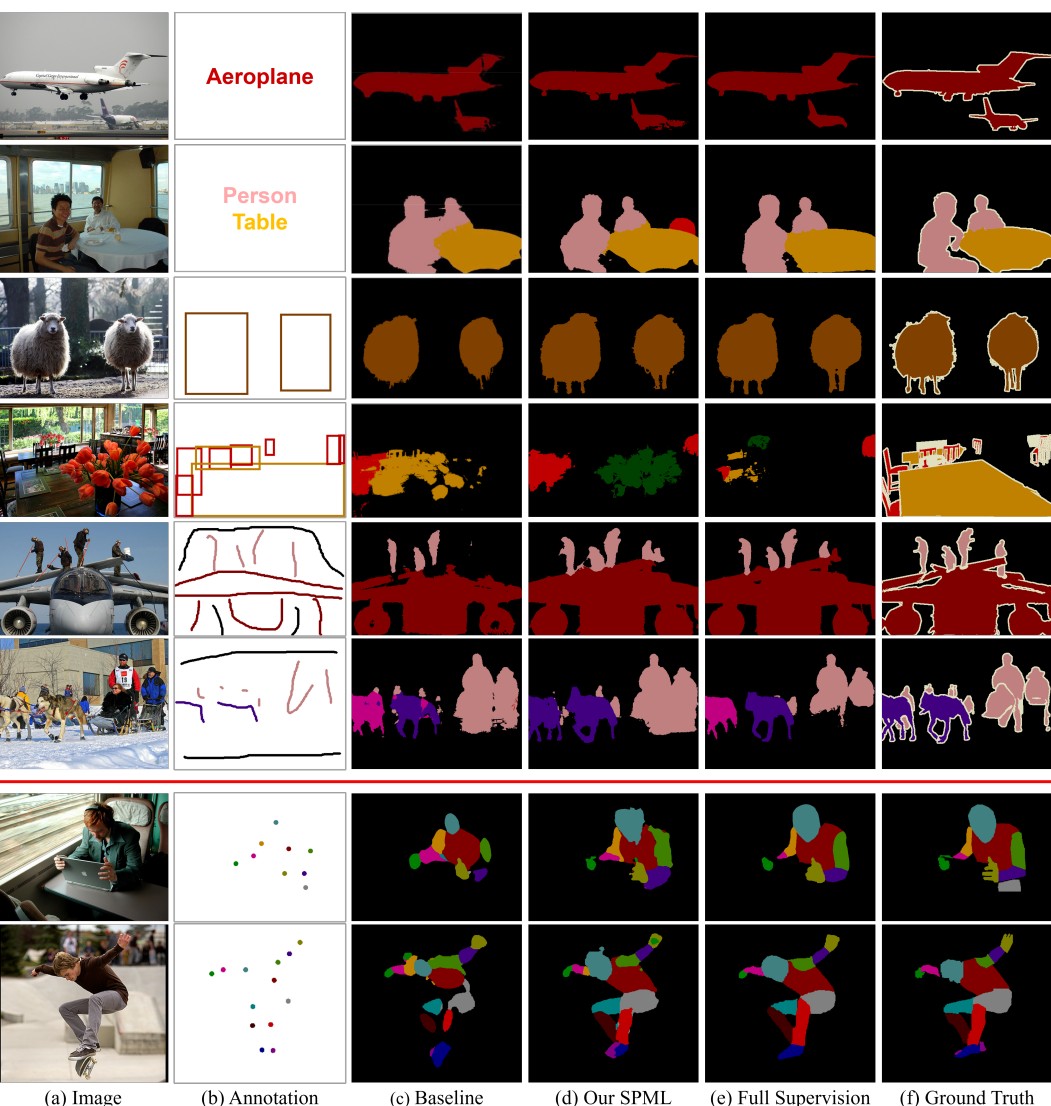

(a) Image (b) Annotation (c) Baseline (d) Our SPML (e) Full Supervision (f) Ground Truth

Figure 8: Visual comparison of baseline method (c), our SPML (d) and fully-supervised SegSort (e) on VOC and DensePose. On VOC (top 6 rows), our baseline method is based on Lee et al. (2019); Song et al. (2019); Tang et al. (2018b) for image tag, bounding box and scribble annotations, respectively. On DensePose (bottom 2 rows), our baseline is Tang et al. (2018b). The results from our weakly-supervised model is visually very close to its fully-supervised counterpart, or even better when visual cues are prominent.

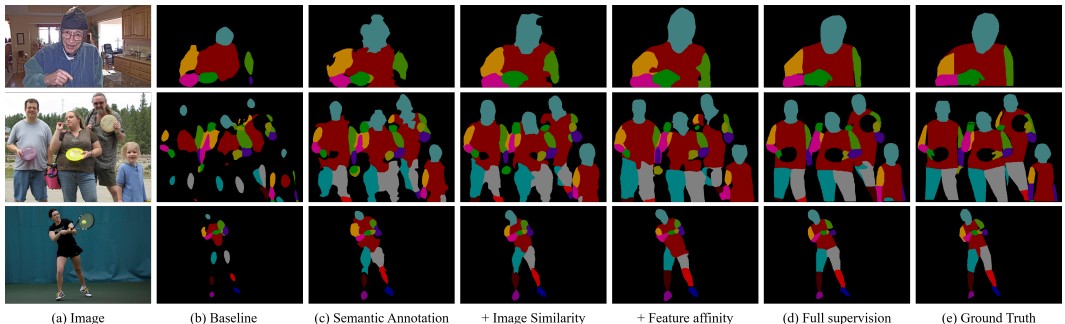

|       |        |        |        |        |        |        |
|-------|--------|--------|--------|--------|--------|--------|
| (a) Image | (b) Baseline | (c) Semantic Annotation | + Image Similarity | + Feature affinity | (d) Full supervision | (e) Ground Truth |

Figure 9: Our segmentation results get better with more types of regularizations. We compare visual results by adding more regularizations. As we introduce more relationships for regularization, we observe significant improvement and our results are visually closer to fully supervised counterparts.

We next describe the hyper-parameters used for each experiment. On Pascal VOC dataset, we set "batchsize" to 12 and 16 for scribble / point and image tag / bounding box annotations. On Dense-Pose dataset, "batchsize" is set to 16. For all the experiments, we train our models with $512 \times 512$ "cropsize". Following Chen et al. (2017), we adopt poly learning rate policy by multiplying base learning rate by $1 - \left(\frac{iter}{max\_iter}\right)^{0.9}$. We set initial learning rate to $0.003$, momentum to $0.9$. For the hyper-parameters in SegSort framework, we use unit-length normalized embedding of dimension $64$ and $32$ on VOC and DensePose, respectively. We iterate K-Means clustering for $10$ iterations and generate $36$ and $144$ clusters on VOC and DensePose dataset. We set the concentration parameter $\kappa$ to different values for **semantic annotation**, **low-level image similarity**, **semantic co-occurrence** and **feature affinity**, respectively. Moreover, $\lambda_I$, $\lambda_O$ and $\lambda_A$ are set to different values according to different types of annotations and datasets. $\lambda_C$ is set to 1 among all the experiments. The detailed hyper-parameter settings are summarized in table 5. We train for $30k$ and $45k$ iterations on VOC and DensePose dataset for all the experiments. We use additional memory banks to cache up previous 2 batches. For conducting experiments, we take advantage of XSEDE infrastructure (Towns et al., 2014) that includes Bridges resources (Nystrom et al., 2015).

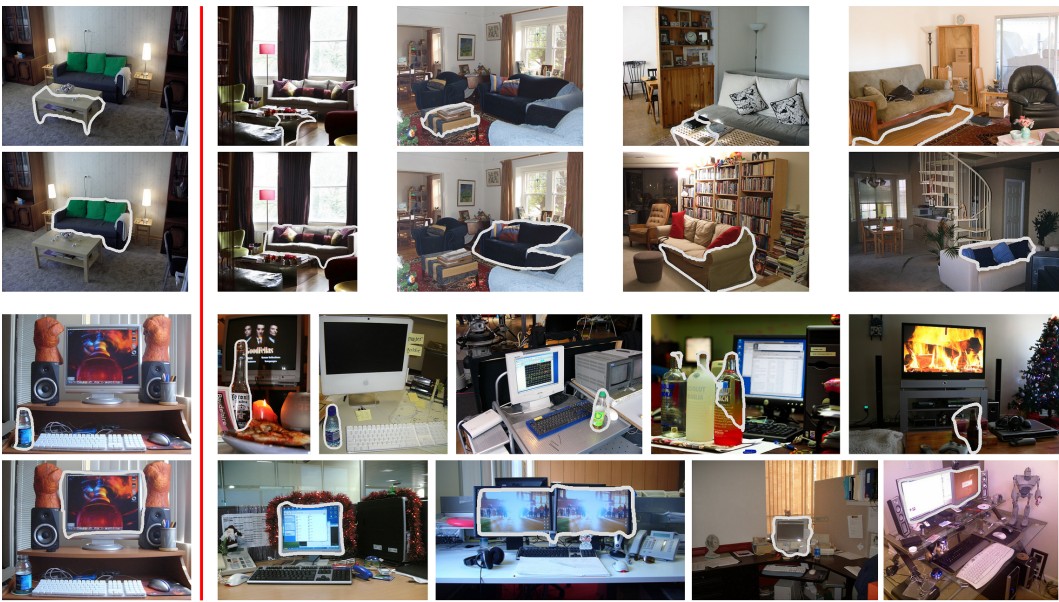

Figure 10: Visual examples of nearest neighbor segment retrievals. We observe that retrieved segments (right) appear in the similar semantic context as the query segments (left). For examples, given a bottle next to a desktop, our model retrieves bottles also next to a desktop.

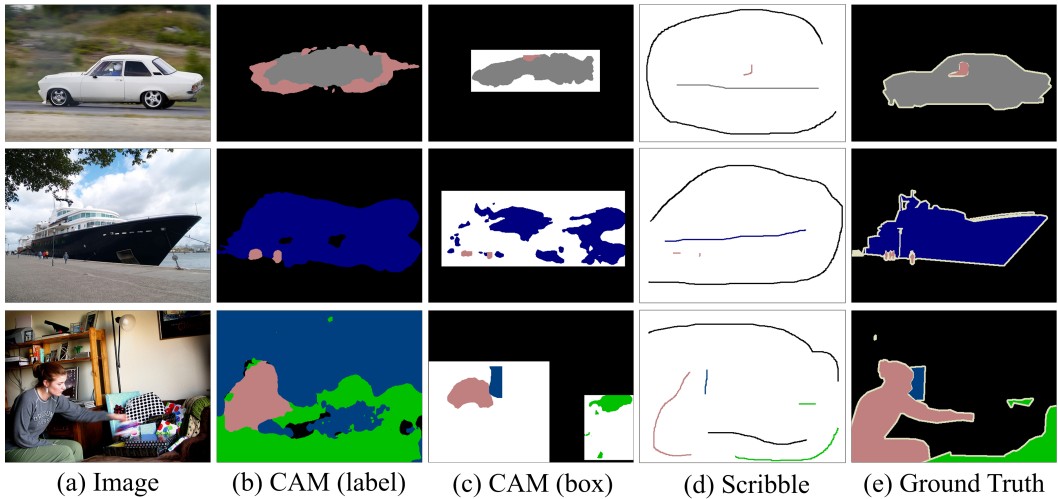

| (a) Image | (b) CAM (label) | (c) CAM (box) | (d) Scribble | (e) Ground Truth |

Figure 11: Visual examples of semantic annotations used on VOC. For image tag and bounding box annotation, we use the classifier trained by Wang et al. (2020) to infer CAM as semantic annotation. These semantic annotations are noisy, which do not precisely localize on the objects.

**Inference and testing.** We fix the learned pixel-wise embedding and train an additional softmax classifier for inference. Iterative training is adopted to bootstrap the semantic segmentation prediction. Notably, we do not propagate gradients to the segmentation CNN from the softmax classifier.

For scribbles / points / bounding boxes, we first learn an initial softmax classifier $S_1$ from the corresponding weak annotations. Following Ahn & Kwak (2018), we apply random walk to refine the semantic logits $\tilde{\mathcal{M}}$ generated by $S_1$. The transition probability matrix $T$ is formulated as follows: $T_{i,j} = (\frac{\exp(\gamma \boldsymbol{e}_i^\top \boldsymbol{e}_j)}{\sum_j \exp(\gamma \boldsymbol{e}_i^\top \boldsymbol{e}_j)})^\beta$, where $\beta$ and $\gamma$ are 20 and 5, respectively. The label propagation is given by: $\tilde{\mathcal{M}}' = T^\top \tilde{\mathcal{M}}$, where $\tilde{\mathcal{M}}'$ denotes refined semantic logits. The random walk process is iterated for 6 times. Next, we obtain the corresponding pseudo labels $\mathcal{Y}_{sc} = \arg\max_c \tilde{\mathcal{M}}'_c$. The pseudo labels are used to train the final softmax classifier $S_2$ for predicting semantic segmentation.

For image tag annotations, we adopt both within-image and across-image label propagation to generate optimal pseudo labels. Starting with CAM logits $\mathcal{M}$, we conduct within-image label propagation thru random walk and obtain refined pseudo labels $\mathcal{Y}_{cam}^1$. Across-image label propagation is carried out by nearest neighbor search thru the whole training set. We refer to SegSort (Hwang et al., 2019) for more details. We then obtain refined pseudo labels $\mathcal{Y}_{nn}^1$ and train the initial softmax classifier $S_1$. Similarly, we use $S_1$ to predict pseudo labels $\mathcal{Y}_{sc}^2$ from the training images. Followed by nearest neighbor search, we obtain our final pseudo labels $\mathcal{Y}_{nn}^2$ and train the final semantic classifier $S_2$.

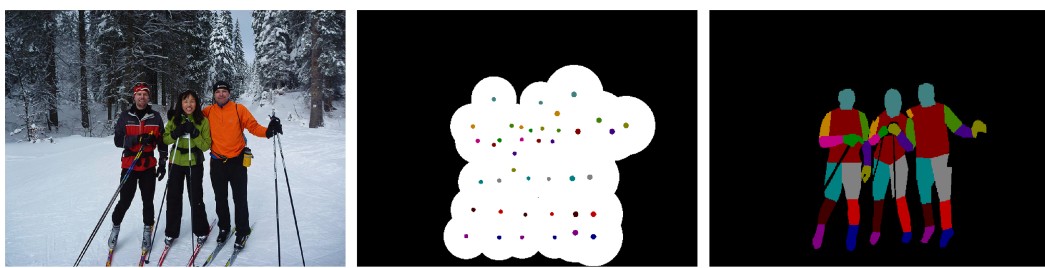

Figure 12: Preparing training labels on DensePose dataset. From left to right are input image, our training labels and ground-truth mask. For each keypoint, a Gaussian heat map is applied to determine labelled, unknown and background region. The white region denotes unknown pixels, to which we propagate labels from annotated or background region.

The inference procedures for different annotations are summarized in algorithm 1 and 2, respectively. For image tags, we adopt multi-scale and horizontally flipping as data augmentation for predicting semantic segmentation. For scribbles / points / bounding boxes, we do not employ data augmentation during the final inference.

| Dataset | Annotation | $\lambda_I$ | $\kappa_I$ | $\lambda_C$ | $\kappa_C$ | $\lambda_O$ | $\kappa_O$ | $\lambda_A$ | $\kappa_A$ | batchsize |
|---------|------------|-------------|------------|-------------|------------|-------------|------------|-------------|------------|-----------|
| VOC | scribbles | 0.1 | 16 | 1.0 | 6 | 0.5 | 12 | 0.0 | - | 12 |
| | points | 1.0 | 16 | 1.0 | 6 | 1.0 | 8 | 0.0 | - | 12 |
| | boxes | 0.3 | 16 | 1.0 | 6 | 1.0 | 8 | 0.0 | - | 16 |
| | image tags | 0.3 | 16 | 1.0 | 6 | 1.0 | 8 | 0.0 | - | 16 |
| DensePose | points | 0.1 | 16 | 1.0 | 6 | 0.0 | - | 0.5 | 12 | 16 |

Table 5: Hyper-parameters for different types of annotations on Pascal and DensePose dataset.

---

**Algorithm 1:** Inference procedure for semantic segmentation using scribble / point / bounding box annotations.

**Input:** Fixed pixel-wise embedding $e$ of the input image and weak annotations $\mathcal{Y}_{weak}$.
**Output:** Semantic segmentation prediction $\mathcal{Y}_{pred}$.
/* Train the initial softmax classifier */
1 Train the softmax classifier $S_1$ using $\mathcal{Y}_{weak}$.
/* Train the final softmax classifier */
2 Predict semantic logits from initial softmax classifier: $\tilde{\mathcal{M}} = S_1(e)$.
3 Calculate pixel-wise transition probability matrix $T$ from $e$.
4 Refine semantic logits by random walk propagation: $\tilde{\mathcal{M}}' = T^\top \circ ... \circ T^\top \tilde{\mathcal{M}}$.
5 Derive pseudo labels from refined semantic logits: $\mathcal{Y}_{sc} = \arg\max_c \tilde{\mathcal{M}}'_c$.
6 Train the softmax classifier $S_2$ using $\mathcal{Y}_{sc}$.
7 Predict final semantic segmentation $\mathcal{Y}_{pred}$ from $S_2$.

---

**Algorithm 2:** Inference procedure for semantic segmentation using image-level tags.

**Input:** Fixed pixel-wise embedding $e$ of the input image and CAM logits $\mathcal{M}$.
**Output:** Semantic segmentation prediction $\mathcal{Y}_{pred}$.
/* Train the initial softmax classifier */
1 Calculate pixel-wise transition probability matrix $T$ from $e$.
2 Refine CAM by random walk propagation: $\mathcal{M}' = T^\top \circ ... \circ T^\top \mathcal{M}$.
3 Derive pseudo labels from refined CAM: $\mathcal{Y}_{cam}^1 = \arg\max_c \mathcal{M}'_c$.
4 Predict new pseudo labels $\mathcal{Y}_{nn}^1$ from $\mathcal{Y}_{cam}^1$ using nearest neighbor retrievals.
5 Train the softmax classifier $S_1$ using $\mathcal{Y}_{nn}^1$.
/* Train the final softmax classifier */
6 Predict pseudo labels $\mathcal{Y}_{sc}^2$ from initial softmax classifier $S_1$.
7 Predict new pseudo labels $\mathcal{Y}_{nn}^2$ from $\mathcal{Y}_{sc}^2$ using nearest neighbor retrievals.
8 Train the softmax classifier $S_2$ using $\mathcal{Y}_{nn}^2$.
9 Predict final semantic segmentation $\mathcal{Y}_{pred}$ from $S_2$.

---

## A.5 ABLATION STUDY OF HYPER-PARAMETERS

We conduct ablation study over different regularizations on Pascal VOC dataset. As shown in table 6, we achieve the most optimal performance on Pascal VOC dataset with $\lambda_I = 0.1$ and $\lambda_O = 0.5$. We also observe performance drops $0.4$ of mIoU by adding **feature affinity** regularization. We argue that scribble/box/point annotations are not uniformly distributed across object instance and background, and results in noisy label propagation.

| $\lambda_I$ | $\lambda_O$ | mIoU |
|---|---|---|
| 0.3 | 0.5 | 73.7 |
| 0.1 | 0.5 | 74.2 |
| 0.05 | 0.5 | 73.5 |
| 0 | 0.5 | 71.7 |

| $\lambda_I$ | $\lambda_O$ | mIoU |
|---|---|---|
| 0.1 | 1.0 | 74.1 |
| 0.1 | 0.5 | 74.2 |
| 0.1 | 0.1 | 74.1 |
| 0.1 | 0 | 72.8 |

| $\lambda_I$ | $\lambda_O$ | $\lambda_A$ | mIoU |
|---|---|---|---|
| 0 | 0 | 0 | 71.2 |
| 0.1 | 0 | 0 | 72.8 |
| 0.1 | 0.5 | 0 | 74.2 |
| 0.1 | 0.5 | 0.1 | 73.8 |

Table 6: Ablation study of different weighting parameters for each objective function on Pascal VOC validation dataset.

| Method | Backbone | CRF | Full | 100% | 80% | 50% | 30% | 0% |
|---|---|---|---|---|---|---|---|---|
| Lin et al. (2016) | DeepLab-MSc-LargeFOV | ✓ | 68.5 | 63.1 | 61.8 | 58.5 | 54.3 | 51.6 |
| Tang et al. (2018b) | DeepLab-MSc-LargeFOV | ✓ | 68.7 | 66.0 | 65.5 | 64.2 | 62.7 | 57.2 |
| Our SPML | DeepLab/ResNet101 | | 76.1 | 74.2 | 74.2 | 73.3 | 73.4 | 71.3 |
| Our SPML | DeepLab/ResNet101 | ✓ | 77.3 | 76.1 | 75.8 | 74.8 | 75.0 | 73.2 |

Table 7: mIoU performance on Pascal VOC 2012 validation set on different lengths of scribble.

| Backbone | aero | bike | bird | boat | bottle | bus | car | cat | chair | cow | table | dog | horse | mbike | person | plant | sheep | sofa | train | tv | mIoU |
|---|---|---|---|---|---|---|---|---|---|---|---|---|---|---|---|---|---|---|---|---|---|
| Tang et al. (2018b) | 83.2 | 35.8 | 82.8 | 66.8 | 75.1 | 90.9 | 83.9 | 89.2 | 35.8 | 82.5 | 53.7 | 83.4 | 83.2 | 79.5 | 82.2 | 57.6 | 81.9 | 41.6 | 81.1 | 73.5 | 73.2 |
| Our SPML | 85.8 | 37.6 | 82.8 | 69.6 | 75.9 | 89.3 | 82.8 | 89.7 | 38.6 | 85.7 | 56.7 | 85.9 | 80.1 | 78.1 | 84.8 | 53.9 | 83.7 | 49.2 | 80.9 | 74.4 | 74.2 |
| Tang et al. (2018b) | 86.2 | 37.3 | 85.5 | 69.4 | 77.8 | 91.7 | 85.1 | 91.2 | 38.8 | 85.1 | 55.5 | 85.6 | 85.8 | 81.7 | 84.1 | 61.4 | 84.3 | 43.1 | 81.4 | 74.2 | 75.2 |
| Our SPML | 89.0 | 38.4 | 86.0 | 72.6 | 77.9 | 90.0 | 83.9 | 91.0 | 40.0 | 88.3 | 57.7 | 87.7 | 82.8 | 79.1 | 86.5 | 57.1 | 87.4 | 50.5 | 81.2 | 76.9 | 76.1 |

Table 8: Per-class results on Pascal VOC 2012 validation set. White- and gray-colored background denotes using without- and with- CRF post-processing for inference.

| Annotations | aero | bike | bird | boat | bottle | bus | car | cat | chair | cow | table | dog | horse | mbike | person | plant | sheep | sofa | train | tv | mIoU |
|---|---|---|---|---|---|---|---|---|---|---|---|---|---|---|---|---|---|---|---|---|---|
| Full mask | 91.5 | 43.5 | 83.0 | 67.9 | 81.7 | 89.8 | 88.7 | 94.6 | 37.5 | 81.6 | 68.7 | 88.8 | 82.4 | 88.6 | 87.6 | 64.1 | 87.6 | 52.7 | 76.5 | 71.4 | 77.3 |
| Scribbles | 87.0 | 36.7 | 82.3 | 65.5 | 79.7 | 89.5 | 84.8 | 90.1 | 37.6 | 86.3 | 63.1 | 89.1 | 87.8 | 83.0 | 86.0 | 65.8 | 85.8 | 60.3 | 76.9 | 73.0 | 76.4 |
| Points | 83.5 | 37.0 | 78.4 | 61.9 | 74.8 | 86.4 | 83.2 | 86.9 | 37.9 | 85.3 | 62.4 | 87.2 | 84.2 | 81.1 | 83.1 | 64.3 | 85.1 | 59.1 | 74.0 | 66.3 | 74.0 |
| Boxes | 84.1 | 36.5 | 86.7 | 57.6 | 75.7 | 87.7 | 84.8 | 89.6 | 39.4 | 86.4 | 57.2 | 89.2 | 88.0 | 82.6 | 80.3 | 54.7 | 88.2 | 55.9 | 79.7 | 71.6 | 74.7 |
| Tags | 82.1 | 38.7 | 80.0 | 56.9 | 73.7 | 85.7 | 81.0 | 86.7 | 33.9 | 87.7 | 60.8 | 86.8 | 84.9 | 81.3 | 77.7 | 53.2 | 86.5 | 50.1 | 64.8 | 58.4 | 71.6 |

Table 9: Per-class results on Pascal VOC 2012 testing set. CRF post-processing is used for inference.

## A.6 MEAN IoU PERFORMANCE WITH VARYING SPARSITY OF SCRIBBLES.

We report absolute mIoU performance by varying sparsity of scribbles on Pascal VOC 2012 validation set. The results are summarized in table 7. Our results are much better with sparser annotation.

## A.7 PER-CATEGORY mIoU ON PASCAL VOC AND DENSEPOSE DATASET.

We next present per-category results on Pascal VOC and Densepose dataset. In table 8, we compare with Tang et al. (2018b) on VOC validation set. Without- and with CRF post-processing, our method outperform the baseline method among most categories by large margin. We further conduct experiments on VOC testing set, using DeepLab as backbone network. In table 9, we can retrieve most performance w.r.t full supervision. We also compare per-category results on DensePose dataset in table 10. We train our baseline method using the code released by Tang et al. (2018b). We outperform the baseline method by large margin in every category.

| Method | bg. | torso | RHand | LHand | LFoot | RFoot | RThigh | LThigh | RLeg | LLeg | LArm | RArm | LFarm | RFarm | Heaad | mIoU | WvF |
|---|---|---|---|---|---|---|---|---|---|---|---|---|---|---|---|---|---|
| Softmax | 96.2 | 73.7 | 61.1 | 57.2 | 37.2 | 37.8 | 56.8 | 54.8 | 49.7 | 49.5 | 62.0 | 63.8 | 58.3 | 61.5 | 84.6 | 60.3 | - |
| SegSort | 95.8 | 71.9 | 57.4 | 53.0 | 33.4 | 33.4 | 54.0 | 51.8 | 46.4 | 46.9 | 59.2 | 61.1 | 54.4 | 57.9 | 83.2 | 57.3 | - |
| Tang et al. (2018b) | 87.2 | 28.3 | 37.5 | 36.0 | 18.9 | 19.5 | 21.2 | 20.8 | 16.1 | 16.6 | 33.9 | 35.3 | 35.6 | 37.6 | 25.2 | 31.3 | 51.9 |
| Our SPML | 93.8 | 57.7 | 48.1 | 43.2 | 22.8 | 22.2 | 36.6 | 35.6 | 27.1 | 27.6 | 42.1 | 45.3 | 42.0 | 45.5 | 72.6 | 44.2 | 77.1 |

Table 10: Per-class results on DensePose minival 2014 set with keypoint annotations. White- and gray-colored background indicates using full and point supervision.

