# OpenReview forum: "Universal Weakly Supervised Segmentation by Pixel-to-Segment Contrastive Learning"
_ICLR.cc/2021/Conference — ICLR 2021 Poster_

### Official Review · AnonReviewer3 · 2020-10-27
**Paper review from AnonReviewer3**

**Rating:** 6
**Confidence:** 5

**Review:**

Summary:

The paper proposes a unified framework for weakly-supervised semantic segmentation that can take various types of weak labels as the input, e.g., points, scribbles, boxes, image tags. The authors formulate it as a contrastive learning framework by considering pixel-to-segment relationships, i.e., for each pixel, finding positive and negative segments to perform contrastive learning objectives. Specifically, the paper introduces four types of relationships between pixels and segments, i.e., image similarity within the image, semantic relationship, semantic co-occurrence, and feature affinity across images. In experiments, results with various weak labels show SOTA performance on the PASCAL and DensePose datasets.

Pros:

The paper is well written and is easy to follow. The proposed unified framework using the idea of pixel-to-segment contrastive learning is interesting for the weakly-supervised semantic segmentation task.

The proposed four types of pixel-to-segment relationships seem mostly reasonable and effective for different types of weak label inputs.

Extensive experiments are conducted to demonstrate the effectiveness of the proposed framework and show strong performance on the PASCAL and DensePose datasets.

Cons:

Although using pixel-to-segment contrastive learning is interesting for weakly-supervised semantic segmentation, the technical contribution is a bit limited as it mostly derives from the recent work in Hwang et al., 2019.

It is not clear to fully understand the effectiveness of the introduced four types of relationships. For example, for semantic co-occurrence, it is doubtful to form positive pairs simply based on the co-occurrence, as the features in two images could be very different even they share at least one same label. This situation could be more significant if performing on more challenging datasets that contain diverse scenes and more categories, e.g., COCO, ADE20K. The authors should make comments on this or provide more analysis for reasoning this design choice, e.g., one easy thing to consider could be weighting this relationship based on the co-occurrence rate.

It appears that feature affinity is less effective or not used in some settings. Although the authors explain the reason in the appendix (due to noisy label propagation), it may not be a principal way to integrate this feature affinity into any dataset. Moreover, Table 1 shows different configurations for different label types, and it could also vary across different datasets. Although there are ablation studies provided in Table 5 and 6 of the appendix, it is difficult to draw conclusions on how to choose the proper weights. In this way, it makes the practical usage harder to scale up to more challenging datasets.

Another concern is the motivation and usefulness of the unified framework. In the literature, there exist specific frameworks optimized for different types of weak labels. Previous frameworks could not use all the weak labels at the same time, which is on the contrary the advantage of the proposed method. Therefore, it is of great interest to see whether the method can simultaneously leverage different weak labels and achieve better performance, e.g., using image tags + boxes against only boxes.

Overall, the paper presents an interesting and effective framework. I would like to hear the feedback from the authors about the above-raised points and may update the rating (my current rating is more like a borderline).

---

> ### Author Response · Authors · 2020-11-24
> **Thanks for acknowledging that our work is interesting, effective, and validated by different benchmarks**
>
> Thanks for acknowledging that our work is interesting, effective, and validated by different benchmarks.  Here we respond to your insightful comments.
>
> Q1: Technical contribution is limited?
> * We are the first one to consider within- and across-image relationships in a contrastive learning formulation in order to provide a unifying solution for various forms of weakly supervised segmentation problems, even though the basic form of the loss was first introduced in SegSort.  More importantly, we showcase how to augment the supervision spatially and semantically by simply mining visual relationships from the data and integrating them in a single, easily optimizable, contrastive learning loss for pixel-wise feature embedding.  While none of these relationships are always correct, they deliver substantial performance gains with the final representation and reach new SOTAs.  The success of our novel modeling approach opens up a new direction on exploring more visual relationships for computer vision tasks in general.
>
> Q2: Effectiveness of the proposed relationships, such as semantic co-occurrence?
> * Yes!  How to capture scene context using semantic co-occurrence relationships can be dependent on the nature of the dataset, and introducing weights by computed categorical co-occurrences is a great suggestion!  For example, if a dataset contains both indoor and outdoor scenes, people appearing with cats and people with vehicles represent two types of scenes and can be modeled separately.
>
> * Furthermore, we’d like to point out that the formulation of our loss function (exp(-d)) also implicitly handles this situation.  So long as in the feature space, there is at least one segment very close to the query pixel, the (pulling) effect of other far away segments would be neglected.  In other words, the learned embedding allows multiple subclusters for one category.  Therefore, if the batch contains enough images to cover at least one other similar scene, this semantic co-occurrence relationship can still be effective.  To conclude, a larger batch size or a memory bank to store cross-batch exemplars could help mitigate this issue for a larger and more complex dataset.
>
>
> Q3: Feature affinity is less effective?
> * We propose 4 kinds of relationships, and not a single one is perfect.  Even the low-level image similarity cue based on detected edges could be undesirable when examined by itself on an individual image.  For example, in Figure 4a, the brown cat is mistakenly attracted to the brown sofa it spread upon, as their colors and textures are highly similar.  What is amazing is that the pixel-wise feature embedding that statistically optimizes such 4 kinds (or sometimes 3 kinds) of noisy pixel-to-segment contrastive relationships could deliver SOTA performance on a variety of weakly supervised semantic segmentation problems with equal ease.
>
> * How generally applicable and how relatively important these 4 relationships (such as feature affinity) are to a specific dataset and annotation type can be decided by their relative weights in a pure data-driven fashion, through hyperparameter selection on the validation set (as done in our experiments).
>
> * Our contribution is to augment supervision by discovering and incorporating various visual relationships into a contrastive learning formulation.  There exist other types of visual relationships that can be explored and utilized pending on the visual task.
>
> Q4: Simultaneously leverage different weak labels?
> * Thanks for the great suggestion!  We conduct the suggested experiments accordingly and plan to include this in the main paper.  We construct two training sets on PASCAL VOC, one with 100% image tags and the other with 50% image tags and 50% bounding boxes.  We train our models and summarize their performance in the table below.  It shows that training with mixed image tags and bounding boxes can improve the performance by almost 2% mIoU.
>
> |   |  100% image tags | 50% image tags + 50% bounding boxes |
> | ------------- |:-------------: |:-------------: |
> | mIoU | 68.6% | 70.3% |

---

### Official Review · AnonReviewer4 · 2020-10-28
**Universal Weakly Supervised Segmentation by Pixel-to-Segment Contrastive Learning**

**Rating:** 6
**Confidence:** 4

**Review:**

In this paper, the authors proposed a metric learning-based semi-supervised semantic segmentation approach. In the proposed method, unlike the conventional semantic segmentation scheme that cast semantic segmentation as pixel-wise classification, they formulate semantic segmentation as pixel-segment contrastive learning. To that end, the authors introduced different positive and negative samples mining mechanisms such as low-level image similarity, semantic annotation, semantic co-occurrence, and feature similarity. Furthermore, the proposed method leverage unlabeled data in discriminative feature learning both within and across images. The validity of the proposed method is demonstrated on Pascal Voc and Dense Pose benchmark datasets.

##########################################################################################

*Strength: The formulation of pixel-to-segment based contrastive learning is intriguing. In addition to that, the authors introduced a new insight into collecting positive and negative samples that would be leveraged in contrastive learning. Moreover, the samples involve pixels that come from both intra and inter images.


#################################################################################

*Weakness:- The sub-modules used in the proposed approaches are adopted from existing methods thus there is no enough novelty in there.
-The resulting improvement achieved is impressive however the work lacks theoretical novelty.

Question: As some of the semantic relationships are basically generating pseudo-labels, for example, Feature Affinity. Have you conducted any experimental analysis to assess the impact of accumulated error that comes from these Pseudo labels?

######################################################################################

*Reason for score:
The proposed work is more of engineering, and it does not have any theoretical novelty; however, formulating semantic segmentation in a contrastive learning context is interesting. Furthermore, the authors demonstrate the effectiveness of the proposed method by showing improvements over the state of the art methods.

---

> ### Author Response · Authors · 2020-11-24
> **Thanks for recognizing that our pixel-to-segment contrastive loss formulation is intriguing and our exploration of relationships within- and across-images is insightful**
>
> Thanks for recognizing that our pixel-to-segment contrastive loss formulation is intriguing and our exploration of relationships within- and across-images is insightful.  Now we address the concerns you raised.
>
> Q1: Lack of theoretical novelty?
> * We are the first one to consider within- and across-image relationships in a contrastive learning formulation in order to provide a unifying solution for various forms of weakly supervised segmentation problems, even though the basic form of the loss was first introduced in SegSort.  More importantly, we showcase how to augment the supervision spatially and semantically by simply mining visual relationships from the data and integrating them in a single, easily optimizable, contrastive learning loss for pixel-wise feature embedding.  While none of these relationships are always correct, they deliver substantial performance gains with the final representation and reach new SOTAs.  The success of our novel modeling approach opens up a new direction on exploring more visual relationships for computer vision tasks in general.
>
> Q2: Assess the accumulated errors from pseudo labels?
> * This is a nice suggested analysis!  We generate pseudo labels using the feature affinity propagation and measure the accuracy using ground truth masks.  We find that such pseudo labels only account for 45.83% accuracy, which is quite inaccurate.  Therefore, using only the feature affinity relationship would deteriorate the training quality.  To conclude, different relationships compensate each other and together enable the model to learn better representations.

---

### Official Review · AnonReviewer1 · 2020-10-28
**WSSS approach which leverages a single pixel-to-segment contrastive learning**

**Rating:** 7
**Confidence:** 3

**Review:**

Summary:
This paper talks about a novel weakly-supervised semantic segmentation (WSSS) approach which leverages a single pixel-to-segment contrastive learning formulation. The key idea is to map each pixel into a point in the feature space so that the pixels in the same semantic categories are embedded closely. It is interesting to note that they have also incorporated the analysis of unlabeled pixels across the images to harvest their patterns/clusters for better discrimination.

Pros:
- Motivation was well described.
- It is interesting to see that four different types of pixel-to-segment (where same segmentation entities were sometimes regarded as different categories) relationships were leveraged in a combined manner to eventually pull up the performance.
- Experiments are reasonably carried out both qualitatively and quantitatively.

Cons/Questions:
- Would there be a reasonable explanation of why a big performance gain was acquired for the "labeled points" case while the jump was relatively small on other weak supervision cases (image tags, bounding boxes, scribbles)?
- A paper by Sun et al. (Mining Cross-Image Semantics for Weakly Supervised Semantic Segmentation, ECCV 2020) also claims that their approach is novel in that they make use of the intra-image information which is somewhat similar to the proposed approach. A paper by Fan et al. (Learning Integral Objects With Intra-Class Discriminator for Weakly-Supervised Semantic Segmentation, CVPR 2020) seems highly relevant to the proposed approach, especially with the CAM-driven WSSS approaches.
Since these recent papers have not been included in the reference of the submitted manuscript, it would be interesting to elaborate on how the proposed approach is unique/strong when compared.
- For Table 2, what is the accuracy measure for the numbers shown?
- It is hard to tell how the lambda values were chosen. (Table 1)
- Some descriptions/explanations are rather redundant and verbose which makes it hard to read.

Typo:
In Section 1, "have have"  --> "have"

---

> ### Author Response · Authors · 2020-11-24
> **We appreciate your recognition that our work is interesting and our results are validated qualitatively and quantitatively**
>
> We appreciate your recognition that our work is interesting and our results are validated qualitatively and quantitatively.  Here we list your questions and answer them accordingly.
>
> Q1: Big performance gain for the "labeled points" annotation type?
>
> * The "labeled points" annotation provides much sparser localization information than scribbles yet more precise foreground localization cues than bounding boxes.  Existing works are highly specialized for either scribbles by propagating and enhancing the mask boundaries or bounding boxes by injecting priors using CAM; neither can generalize well to the more extreme "labeled points" annotations.  Our method unifies all these types of weak supervision with equal ease and is thus able to provide most performance gain for this challenging annotation type.
>
>
> Q2: Compare with Sun et al. 2020 and Fan et al. 2020?
>
> * Thanks for referring to these two works!  We will introduce them in the related work as well.  [Sun et al 2020] considers within-image relationships and explores the idea of co-segmentation, whereas [Fan et al 2020] estimates the foreground and background for each category, with which the network learns to generate more precise CAMs.   Our method approaches weakly supervised semantic segmentation as a semi-supervised metric learning problem, with the goal of propagating semantic labels to unlabeled pixels in a unifying and flexible contrastive learning framework.  A unique, novel, and most advantageous aspect of our method is that we contrast both labeled and unlabeled pixels within and across images with equal ease, bringing a substantial performance gain that has not been met by any other approach.
>
> Q3: The metric in Table 2?
> * The metric in Table 2 is based on mIoU, the most commonly adopted semantic segmentation metric.
>
> Q4: How to decide lambda values in Table 1?
> * The lambda values are decided by experimental validation, as a generic hyperparameter selection problem.  Table 5 in the Appendix details our ablation studies on the effects of loss-weighting parameters.
>
> Q5: Descriptions can be crisper?
> * Yes!  We’ll keep polishing the writing of our paper.

---

### Official Review · AnonReviewer2 · 2020-10-29
**great performance but a bit incremental**

**Rating:** 5
**Confidence:** 4

**Review:**

The submission proposes a unified framework for weakly supervised semantic segmentation which is compatible with different types of annotations including image tag, bounding box, points and scribbles. With different kinds of labels, the method derives positive and negative segments for each labels and via metric learning the network learns to predict class label for each pixel. While the proposed method obtained STOA or close to SOTA performance on VOC2012 and densepose dataset, the reviewer feels that the novelty of the proposed method is not significant enough. A unified framework for semantic segmentation is appealing but it isn't new. E.g. Guided Attention Inference Network, TPAMI 2019 proposed a way to use gradient-based attention weakly supervised semantic segmentation framework which is compatible with image tags, bounding boxes and pixel-level annotations. In their work, different kinds of annotations are converted into supervision on class-specific attention maps and no parameters need to be tuned based on the ratio/types of annotations are used in the training, whereas the proposed method here needs different parameter choices for different lables/dataset. In the reviewer's eyes the submission follows a similar direction, although the framework design and objectives are different. The reviewer hope the authors can clarify their contribution and add discussions comparing to other similar approaches.

---

> ### Author Response · Authors · 2020-11-24
> **Thank you for acknowledging that our proposed framework unifies various annotations for weakly supervised semantic segmentation and achieves SOTA results**
>
> Thank you for acknowledging that our proposed framework unifies various annotations for weakly supervised semantic segmentation and achieves SOTA results.
>
> Q: Not new compared to Guided Attention Inference Network (GAIN)?
>
> * Thanks for bringing this work to our attention!  We will also mention this comparison in the main paper.  To compare our method with GAIN, we would like to first point out that GAIN only deals with image tags, bounding boxes, or full segmentation masks.  The idea of GAIN is to utilize these annotations to refine class-specific attentional maps.  On the other hand, our framework is more general, with which we tackle all types of weak annotations, including scribbles, points, images tags, and bounding boxes.
>
> * It is worth noting that each of the annotations carries different assumptions: 1) Image tags provide only semantic information but no locations; 2) Bounding boxes precisely indicate background regions, but only roughly outline the size and location of objects, without knowing the pixels on the objects; 3) Scribbles and points show the object foreground, but the boundary between background and foreground remains unknown.
>
> * In particular, it is unclear how GAIN can be  extended to scribbles or points, which provide more precisely localized cues than other forms of weak annotations.  In contrast, our method unifies all these types of weak annotations (each with its own characterization and assumptions) in a single contrastive learning framework and achieves SOTA performance across the board.

---

### Official Review · AnonReviewer5 · 2020-11-07
**The paper performs pixel to segment metric learning based on various heuristics and obtains clear SOTA results on various weakly-supervised visual learning tasks**

**Rating:** 7
**Confidence:** 5

**Review:**


Weakly-supervised image/object segmentation can naturally be formulated as a pixel-level semi-supervised problem. To solve the insufficient supervision problem in weakly-supervised segmentation, the paper proposes to use four kinds of heuristics to provide supervision for training pixel-level metric learning. The takes the advantages of image segments generated from edge detectors (HED & gPB), then low-level image similarity, semantic annotation, semantic co-occurrence, and feature affinity.

The paper has obvious advantages summarized as follows.
1. It unifies various weakly-supervised segmentation tasks by proposing pixel-level metric learning.
2. The paper obtains clear SOTA results and significantly promotes the development of weakly-supervised segmentation.

The weakness of the paper is its clarity. Lots of details are missing in the paper, e.g., I cannot find how to apply semantic co-occurrence applied in training segmentation models using image classification models. I think this is not the authors' fault. The problem is caused due to there are so many contents in the paper. Thus, I think source codes should be released and well-organized to ensure the reproducibility of the paper.

Besides, the terms in the papers can be re-considered. (1)  Although contrastive learning is very popular now, I think the method is better termed as metric learning - its original name. (2) I think there are some redundancy in semantic annotation, semantic co-occurrence, and feature affinity. The relation between them should be clarified and justified.

---

> ### Author Response · Authors · 2020-11-24
> **Thank you for recognizing our efforts in unifying various annotation types in a metric learning framework and our SOTA performance across all annotations**
>
> Thank you for recognizing our efforts in unifying various annotation types in a metric learning framework and our SOTA performance across all annotations.  Here we list your questions and provide our answers.
>
> Q1: How to apply semantic co-occurrence using image classification models?
>
> * For semantic co-occurrence, we consider whether an image contains a specific object class. For a segment in a training image, all the segments in any other images that contain the same category are positive examples, or otherwise negative. We can thus mine the semantic co-occurrence relationship from either pixel labels, bounding boxes or image tags. The image classification model is not required for the relationship.
>
> Q2: Release code?
>
> * Yes, we are also committed to open source research and will release our code upon publication.
>
> Q3: Metric learning vs contrastive learning?
>
> * While we agree that metric learning is a more general term, our method emphasizes on contrasting segments based on different relationships.  Our objective is to contrast a pixel against segments other than its own. Therefore, we name the title of our paper in terms of pixel-to-segment contrastive learning to highlight such a perspective.
>
> Q4: Clarification on the relations between semantic annotations, semantic co-occurrences, and feature affinities?
>
> * Semantic annotations, feature affinities, and semantic co-occurrences characterize semantic relationships at an increasingly higher level:  1) Our supervision is first based on sparsely labeled pixels (scribbles / points / CAM); 2) We then augment our supervision by propagating labels to unlabelled pixels in the feature space, via the feature affinity relationship; 3) We finally consider more global regularization at the scene level, in terms of semantic co-occurrence relationships.  These relationships together provide pseudo supervision from low- to high- semantic levels, and from local to more global spatial coverage.

---

### Public Comment · ~Wanxuan_Lu1 · 2020-11-13
**The inference phase requires pixel-level annotations of the training set and takes a long time?**

In Appendix A.4, the authors said, "We produce oversegmentations and calculate the prototypes in the training set. For Pascal VOC and DensePose dataset, We partition each image into 144 and 576 clusters, respectively. We set k = 20 for nearest neighbor retrievals, and then, we infer the semantic label of each queried segment with majority label of retrievals."

I have two questions about this inference operation. First, "label of retrievals" means that the pixel-level annotations of the training set are required?  Second, this inference operation takes a long time? For DensePose, each image will be partitioned into 576 clusters, and retrieval must be performed 576 times for each image. Over-segmentation and retrieval are both time-consuming operations.

---

> ### Author Response · Authors · 2020-11-24
> **Thanks for your thoughtful questions on our inference pipeline**
>
> Thanks for your thoughtful questions on our inference pipeline!
>
> Q1: Are pixel-level annotations of the training set required?
> * We infer the segment labels in the training set by the sparsely labeled set.  After deciding the label of each segment in the training set, we use them for the nearest neighbor search to predict the label of query segments during inference.  We’ll include detailed description in the final version.
>
> Q2: Does the inference take a long time?
> * We refer to Table 4 in the SegSort paper for the ablation study on runtime.  The computation bottleneck is the iterative K-Means algorithm rather than nearest neighbor search as the latter can be well parallelized on GPUs.  Also, one can replace the nearest neighbor search by training a linear classifier.  In terms of the runtime of iterative K-Means for 576 segments, one can consider each iteration is equivalent to a convolutional layer with 576 dimension on top of the last embedding layer, whose resolution is very small.  Therefore, the runtime is still manageable as reported in the SegSort paper.

---

### Decision · Program_Chairs · 2021-01-07
**Final Decision**

**Decision:**

Accept (Poster)

**Comment:**

This article proposes a novel weakly supervised segmentation method that unifies several annotation types using contractive/metric learning. This method clearly outperforms the current SOTA. While the unified framework itself is not novel enough, the reviewers agree that the contrastive loss formulation is interesting and the extensive experiments show its effectiveness. Overall, I consider that this unified framework is well engineered, the formulations are insightful, and the results advance the SOTA of weakly supervised segmentation. Accordingly, I propose to accept this paper at ICLR 2021.